# LncRNA-Mediated Tissue-Specific Plastic Responses to Salinity Changes in Oysters

**DOI:** 10.3390/ijms26104523

**Published:** 2025-05-09

**Authors:** Mengshi Zhang, Jinlong Zhao, Ao Li, Mingjie Zhao, Meitong Huo, Jinhe Deng, Luping Wang, Wei Wang, Guofan Zhang, Li Li

**Affiliations:** 1College of Life Sciences, School of Marine Science and Engineering, Qingdao Agricultural University, Qingdao 266109, China; zhangms@qdio.ac.cn (M.Z.); zhaojl@qdio.ac.cn (J.Z.); 2State Key Laboratory of Breeding Biotechnology and Sustainable Aquaculture, Institute of Oceanology, Chinese Academy of Sciences, Qingdao 266000, China; gfzhang@qdio.ac.cn; 3Laboratory for Marine Biology and Biotechnology, Qingdao Marine Science and Technology Center, Qingdao 266237, China; 4Key Laboratory of Experimental Marine Biology, Institute of Oceanology, Chinese Academy of Sciences, Qingdao 266071, China; wanglp@qdio.ac.cn (L.W.); wangwei@qdio.ac.cn (W.W.); 5Oyster Industrial Technology Institute of Zhanjiang, Southern Marine Science and Engineering Guangdong Laboratory (Zhanjiang), Zhanjiang 524000, China; 6National and Local Joint Engineering Laboratory of Ecological Mariculture, Qingdao 266071, China; 7University of Chinese Academy of Sciences, Beijing 100049, China; zhaomj@qdio.ac.cn (M.Z.); huomt@qdio.ac.cn (M.H.); dengjh@qdio.ac.cn (J.D.); 8Laboratory for Marine Fisheries Science and Food Production Processes, Qingdao Marine Science and Technology Center, Qingdao 266071, China

**Keywords:** estuary, RNA-seq, gene expression regulation, stress response, shellfish

## Abstract

Salinity is a key environmental factor influencing the survival of aquatic organisms, and transcriptional plasticity is a crucial emergency response to environmental changes. However, most transcriptomic studies on salinity responses have not explored the expression patterns and regulatory mechanisms across different tissues. The Suminoe oyster (*Crassostrea ariakensis*), a sessile estuarine species that inhabits fluctuating salinity environments, provides an excellent model for studying the molecular basis of salinity response divergence. All eight tissues responded to acute salinity stresses and exhibited distinct tissue-specific expression patterns in both mRNA and long non-coding RNA (lncRNA) profiles across three salinity conditions. The hepatopancreas and striated muscle were identified as tissues specifically sensitive to hyper- and hypo-saline stress, respectively, based on the number, expression pattern, and plasticity of differentially expressed genes (DEGs). We established lncRNA-mRNA regulatory relationships that environmentally responsive lncRNAs enhanced DEGs’ expression and underpinning tissue-specific responses. Under moderate stress, the hepatopancreas and striated muscle initiated positive responses related to water transport and shell closure, respectively. Under severe stress, the hepatopancreas activated cellular resistance pathways, while the striated muscle experienced significant cell death. Our findings provide insights into lncRNA-mediated, tissue-specific environmental responses and lay the foundation for further research into the adaptive evolution of tissue-specific regulation.

## 1. Introduction

Organisms are subject to a variety of abiotic selective pressures in their lifecycle, among which salinity is one of the most important environmental factors for those living in the aquatic environment [1]. Coastal ecosystems, especially in estuarine regions, are rendered more susceptible to salinity fluctuations as a result of the effects of tides, freshwater input, evaporation, and the increasing presence of anthropogenic pollutants [2]. Differences in salinity across estuarine regions limit the survival and distribution of marine organisms; therefore, studying the salinity responses and adaptation of marine organisms is of great significance [3]. Marine organisms undergo morphological, physiological, and molecular changes to respond or adapt to environmental changes [4]. While organisms can adapt to these stresses by means of genetic variations through long-term evolution from generation to generation, migrating to suitable environments elsewhere and phenotypic plasticity without altering genetic components are two basic ways to respond to environmental disturbances [5,6,7]. Phenotypic plasticity refers to the ability of a single genotype to produce different phenotypes under different conditions [8]. Plastic response is particularly important for sessile organisms that cannot move for coping with challenging conditions [9,10,11]. Phenotypic plasticity can serve as an emergency response, irrespective of the fitness, by allowing adjustments in physiological properties, which are often necessary for organismal survival when exposed to stressful environments [12,13]. Therefore, assessing the plastic capacity to salinity fluctuations of sessile organisms dwelling in estuarine zones and its underlying mechanisms is of significant theoretical and practical importance.

Oysters are representative sessile bivalve mollusks that inhabit intertidal zones, estuaries, and shallow marine environments, serving as a model species in evolutionary and ecological studies [14]. Almost all of *Crassostrea* oysters have evolved euryhaline habits, especially for those inhabiting estuarine environments, to cope with fluctuating salinity conditions. Several studies have evaluated the morphological, physiological, proteomic, transcriptional, and genetic processes underpinning short-term plastic responses and long-term evolutionary adaptation to salinity changes in different oyster species at various developmental stages [11,15,16,17,18,19,20]. Common biological processes have been found to be associated with salinity responses and adaptation across oyster species and lifecycles, including changes in the regulation of osmotic agents (e.g., inorganic ions, free amino acids, transporters, and other solutes), altered energy metabolisms (e.g., glycolysis, the tricarboxylic acid (TCA) cycle, pyruvate metabolism), and changes in stress and immune responses [21,22,23,24,25,26,27]. Transcriptional studies have revealed that there is no common set of genes in these processes across these oyster species. Rather, oyster species usually use different genes, although they belong to similar functional categories, to respond to/adapt to salinity challenges [21,22,26,27].

The Suminoe oyster (*Crassostrea ariakensis*) is broadly distributed in estuaries of East Asia over wide ranges of latitude and experiences various salinity gradients [28,29]. Previous studies demonstrated that different geographic populations exhibited divergent response capacities to salinity stresses at both juvenile and adult stages [9,11,19,23]. Using whole-genome sequencing, significant genomic signatures (e.g., single nucleotide polymorphisms (SNPs), gene expansions, three dimensional structures) that potentially affect the oyster’s responses to salinity changes were identified [9,11,15,19]. Substantial research has focused on the gene expression patterns of *C. ariakensis* associated with their responses to salinity challenges [21,22,23]. However, the full range of variation in transcriptional processes underpinning the salinity responses has not been evaluated, since studies largely focus on whole-body or single-tissue expression patterns: within the same individual, different tissues may span the range from quiescent to active.

Tissue specificity of gene expression widely exists across terrestrial and aquatic animals that always associated with divergent functions of different tissues [30]. The key physiological basis is the tissue-specific expression patterns of genes and endogenous metabolites, which subsequently regulate the tissue’s function and response to environmental changes. Genes and metabolites responsible for shell closure have evolved divergent expression patterns that are specifically highly expressed in certain tissues, such as class II alpha-mannosidase and mannose-containing glycan, showed high expression levels in the smooth muscle that controls the shell closure of the oyster [9]. Different tissues also showed distinct environmental sensitivity, for example, the gill of a fish and the digestive gland of the oyster are specifically enriched for trace metals and foodborne norovirus, respectively [31,32]. Since all tissues share the same genomic components, the regulation of genome-wide gene expression is crucial for the generation of specific patterns across tissue types [33]. Recent advances in genomic and epigenetic sequencing provide a good opportunity to deeply explore regulatory mechanisms underlying tissue-specific transcriptional profiles.

Genomic variations (e.g., promoter and enhancer sequences, duplications) and epigenomic modifications (e.g., long non-coding RNA (lncRNA), histone modification, three dimensional genome structure) are two fundamental mechanisms that affect tissue-specific transcriptional profiles [34,35,36,37]. Due to tissue-specific, development-specific, and condition-specific expression patterns [38,39,40,41], lncRNAs, longer than 200 nucleotides in length, have attracted more attention for their critical roles in transcription regulation [42,43]. LncRNAs are not translated into proteins but instead function intrinsically as RNA molecules. Functions of lncRNAs are linked with their specific subcellular localizations and their specific interactions with DNA, RNA, and proteins. LncRNAs can alter the translation of mRNAs [44,45] and interfere with signaling pathways, such as embryonic development [46], cell differentiation [47], and disease progression [48]. Generally, lncRNAs regulate adjacent or associated genes via *cis*- or *trans*-acting mechanisms, tuning downstream physiological and biochemical reactions in response to environmental changes [49,50,51]. Notably, *cis*-regulation of lncRNAs on nearby genes, such as acting as enhancers, is more common in most species when encountered with biotic and abiotic stresses [9,51,52,53]. In oysters, lncRNAs can modulate gametogenesis by regulating sterility-specific genes [54], control shell color by regulating pigment genes [55], and respond to heat stress by regulating chaperone (HSP) genes [56]. Furthermore, lncRNAs play important roles in molluscan biomineralization [57,58,59]. Previous studies have identified novel biomineralization-related candidate genes and shell formation-regulating lncRNAs, providing evidence that lncRNAs regulate shell growth and shape in bivalves [57]. However, little is known about genome-wide tissue-specific expression patterns and their interactions with the mRNAs of lncRNAs in oysters’ plastic responses.

In this study, we evaluated and compared the mRNA and lncRNA profiles of eight tissues (gill [G], smooth muscle [SmM], striated muscle [StM], marginal mantle [MM], inside mantle [IM], heart, lip, and hepatopancreas [HE]) in the Suminoe oyster in response to hyper- and hypo-saline stresses. We found that *C. ariakensis* exhibited clearly tissue-specific responses to salinity stresses at both mRNAs and lncRNAs levels, with HE and StM identified as the most sensitive tissues in response to hyper- and hypo-saline conditions, respectively. Environmentally responsive lncRNAs played an important role in enhancing the expression of associated DEGs and recruiting divergent biological processes upon salinity stresses between these two tissues. Our findings revealed lncRNA-mediated tissue-specific environmental responses in marine invertebrates, highlighting the significance of tissue-specific assessments for understanding both the mechanisms and the evolution of short-term responses and long-term evolutionary adaptation.

## 2. Results

### 2.1. Differences in the Response to Salinity Among the Eight Tissues

Both G and lip showed higher gene expression compared with other tissues under three salinity conditions, while both SmM and heart showed a lower gene expression under three salinity conditions (*p* < 0.05, Figure 1a and Appendix A). The MM and HE showed a higher expression of lncRNAs under normal environment (20‰) (*p* < 0.05, Figure 1b). The HE and StM showed a higher expression of lncRNAs under hypo-saline condition (0‰), while HE and heart showed a higher expression of lncRNAs under hyper-saline condition (50‰) (*p* < 0.05, Appendix A). The expression level of lncRNAs was significantly higher than mRNAs in all of eight tissues regardless of salinity conditions (*p* < 0.01, Figure 1c and Appendix A).

All eight tissues responded to salinity stresses that both genome-wide expression profiles of mRNAs and lncRNAs were independently clustered between normal and stressed conditions (Figure 2a,b and Appendix A). Although some tissues were clustered together at each of the three salinity conditions, genome-wide expression profiles of mRNAs among StM, HE, G, lip, and heart showed divergent patterns, which was clearly more than those of lncRNAs (Figure 2c and Appendix A).

### 2.2. DEGs and DELRs in Response to Hyper- and Hypo-Saline Stresses

A total of 1883 and 1973 genes in lip and StM were identified as DEGs in response to salinity stresses (hyper- and hypo-saline stresses), while up to 2494 and 2487 DEGs were identified in SmM and HE (Appendix A). Under hypo-saline condition, only 1001 and 1033 genes in lip and StM were identified as DEGs, while up to 1692 and 1891 DEGs were identified in SmM and HE (Figure 3a). Under hyper-saline condition, only 932 and 1049 genes in HE and MM were identified as DEGs, while up to 1369 and 1334 DEGs were identified in StM and G (Figure 3b).

Only 9 and up to 32 DELRs were identified in IM and G in response to salinity stresses, respectively (Appendix A). Under hypo-saline condition, only 2 and up to 12 of DELRs were identified in StM and SmM, respectively (Figure 3c). Under hyper-saline condition, only 8 and up to 28 of DELRs were identified in IM/heart and G, respectively (Figure 3d).

The number of DEGs upon hyper- and hypo-saline stresses was slightly negatively correlated among eight tissues (*ρ* = −0.67, *p* = 0.070, Figure 3e), while the number of DELRs upon hyper- and hypo-saline stresses showed no correlation among the eight tissues (*ρ* = 0.013, *p* = 0.98, Figure 3f).

Under salinity stress, all DEGs in StM, lip, HE, and SmM were significantly enriched in the stress-response related terms of “response to stress” (HE: *DNAJB13*, lip: *STI1*, SmM: *STI1*, StM: *STI1*) and “superoxide metabolic process” (HE: *Dominin*, lip: *Dominin*, SmM: *Dominin*, StM: *Dominin*) (Appendix A). Additionally, in HE, the salinity-response related terms were enriched in “glutamine biosynthetic process” (HE: *GS*); in lip, they were enriched in “oxidation−reduction process” (lip: *FMO*) and “response to stimulus” (lip: *TLR6*); and in SmM and StM, they were enriched in “oxidation−reduction process” (SmM: *SDH*, StM: *FMO*) (Appendix A).

### 2.3. Transcriptional Changes in Genome-Wide Genes and DEGs

For genome-wide genes, HE and SmM exhibited significant higher transcriptional plasticity (*p* < 0.05, HE: 0.314 [median value], SmM: 0.313), while lip, heart, and StM showed significant lower transcriptional plasticity under hypo-saline conditions (*p* < 0.05, lip: 0.21, heart: 0.21, StM: 0.22, Figure 4a). Under hyper-saline conditions, SmM exhibited significant higher transcriptional plasticity (*p* < 0.05, SmM: 0.22), while MM showed significant lower transcriptional plasticity (*p* < 0.05, MM: 0.17, Figure 4b). For DEGs of each tissue, both HE and StM showed significant lower transcriptional plasticity under both hyper- (HE: 0.40, StM: 0.50) and hypo-saline (HE: 0.77, StM: 0.64) conditions (*p* < 0.05, Figure 4c,d). DEGs in StM and HE exhibited significant lower expression levels under hypo- and hyper-saline conditions, respectively (*p* < 0.05, Appendix A). Considering that the expression patterns and plastic capacity between StM and HE were different with other tissues in respond to salinity challenges, we focused on StM and HE in subsequent analysis.

### 2.4. Expression Patterns of mRNAs and lncRNAs in HE and StM

Under hypo-saline conditions, six and two lncRNAs in HE and StM were identified as DELRs, and the number of DEGs significantly associated with them were 767 and 336. Under hyper-saline conditions, 16 and 21 lncRNAs in HE and StM were identified as DELRs, and the number of DEGs significantly associated with them were 628 and 987 (Table 1 and Appendix A).

The expression level of DEGs associated with DELRs was higher than that of DEGs unrelated to DELRs in both HE and StM in response to salinity stresses (*p* < 0.01, Figure 5a). Also, for non-differentially expressed genes (nDEGs), expression levels of nDEGs associated with DELRs were higher than that of nDEGs unrelated to DELRs in both HE and StM in response to salinity stresses, especially under hypo-saline conditions (*p* < 0.01, Appendix A).

Expression levels of *CREBL2* and *MSTRG.46263.1* were both significantly increased in response to hyper-saline stress (*p* < 0.05, Figure 5b). We noticed that DEGs of *CREBL2* (cAMP-responsive element-binding protein-like 2) and associated DELRs (*MSTRG.46263.1*) are closely located at the chromosome 10 (34 kb). The upstream region (7 kb) of *CREBL2* and *MSTRG.46263.1* were located in accessible regions, and they interacted with each other via HiC-loops. Moreover, both Hi-C interaction and chromatin accessibility were enhanced when the oysters were translocated from the native north environment to a non-native south environment (Figure 5c).

### 2.5. Biological Processes of HE and StM upon Hypo- and Hyper-Saline Conditions

Under hypo-saline conditions, significantly enriched terms of DEGs associated with DELRs in StM were responses to stress (heat shock proteins: *HSP68*, *HSP70A*, *HSP70B*; *STI1A*: stress-induced protein 1A; *GADD45*: growth arrest and DNA-damage-inducible protein 45), programmed cell death (*IAPs*: inhibitors of apoptosis) and so on (*p* < 0.01, Figure 6a). In HE, the term of water transport was significantly enriched (*AQP8*: aquaporin 8) (*p* < 0.01, Figure 6b).

Under hyper-saline condition, significantly enriched terms in StM included mannan catabolic process (*β-mannanase*: mannan endo-1,4-beta-mannosidase) and the ubiquinone biosynthetic process (*COQ7*: ubiquinone biosynthesis protein COQ7-like protein) (*p* < 0.01, Figure 6c). In HE, significantly enriched terms included glutamine and proline biosynthetic process (*GS*: glutamine synthetase; *PYCR1*: pyrroline-5-carboxylate reductase 1), response to stress (*STI1B* and *STI1C*), and cell–matrix adhesion (*XBP1*: X-box binding protein-like protein 1) (*p* < 0.01, Figure 6d). DEGs of enriched terms showed significantly higher expression under salinity stresses than that of normal condition, except for *XBP1* (Figure 6e).

## 3. Discussion

Our results demonstrated that all eight tissues of the Suminoe oyster in this study responded to acute salinity changes in both the mRNA and lncRNA profiles. There were clearly tissue-specific expression patterns especially for divergent mRNA profiles among eight tissues at each salinity conditions, while there were weak tissue-specific expression patterns for lncRNAs profiles. The striated muscle and the hepatopancreas were identified as two sensitive tissues that highly responded to hyper- and hypo-saline conditions. DEGs associated with environmentally responsive lncRNAs (DELRs) increased their expression levels upon salinity stresses, indicating regulatory functions as enhancers for lncRNAs in oysters, which is also revealed in model species, such as *Mus musculus* [60]. Biological processes of DEGs associated with DELRs corresponded to divergent categories relating to salinity responses, which may result from the differential functions between these two tissues in oysters.

Rather than evaluating the salinity responses at whole-body or single tissue levels in oysters [2,21,61], we found that there was tissue-specific divergence at genome-wide gene expression profiles upon three salinity conditions, indicating differential transcriptional responses to salinity changes across multiple tissues. The expression profiles of lncRNAs also exhibited clearly tissue-specific patterns at the normal condition, and several tissues also showed divergent transcriptional patterns at hyper- or hypo-saline conditions. Since previous studies largely focused on divergent adaptation or responses to salinity gradients/changes among different species [21,22] or populations [9,11,62] inhabiting diverse salinity environments, as well as salinity-adapted variations at the organism level [16,19]. Our results highlighted that tissue-specific environmental responses are also significant and pervasive in marine species, in addition to model species such as humans, mice, and *Arabidopsis thaliana* [63,64,65]. Moreover, according to the number of DEGs, tissues, such as striated muscle and gill, showing widespread responses to hyper-saline stress (more DEGs) exhibited limited responses to hypo-saline stress (less DEGs), and vice versa (hepatopancreas and smooth muscle showed more DEGs to hypo-saline stress but less DEGs to hyper-saline stress). While not significant, the negative relationship between hyper-saline and hypo-saline responses indicated that the oyster has evolved a trade-off strategy among emergency responses of tissues to different stressors. This pattern is consistent with reports in other organisms, such as sugarcane and red algae, where resource allocation for stress responses involves prioritization and trade-offs under varying environmental conditions [66,67]. Evolutionary trade-offs are commonly recruited in oysters and other marine species at several hierarchical levels: growth and stress tolerance [7,21,68,69,70], the contents among energetic metabolites [7,10,70,71], baseline and plastic expression level of genes [72,73], and genomic and epigenetic variations for transcriptional plasticity [9]. Our findings stressed the importance of tissue specificity that should be comprehensively considered in future studies on long-term adaptation and contemporary response to changing environments.

Except for differential responses to hyper- and hypo-saline stresses in terms of PCoA clustering and the number of DEGs, there was considerable divergence in expression patterns and plastic capacity between striated muscle and hepatopancreas in respond to salinity challenges. Under hypo-saline stress, the hepatopancreas exhibited higher DEG induction but lower transcriptional plasticity, contrasting with responses under hyper-saline conditions. The striated muscle showed the opposite pattern. Due to strong plastic response would result in high energy cost [74,75,76], organisms prefer to activate more genes but with minor effects (plastic changes) at the initial phase of or temperate environmental stresses, which is a common strategy for oysters when exposed to challenging conditions [7,10,70]. Thus, we proposed that the hepatopancreas was more sensitive (weak tolerance) to hyper-saline stress, while the striated muscle was more sensitive to hypo-saline stress. Such conspicuously tissue-specific divergent responses are tightly linked to their distinctive biological functions. Hepatopancreas possess higher antioxidative capacity showing elevated enzyme activities of superoxide dismutase and catalase, and increased gene expression of *metallothionein*, which substantially detoxified reactive oxygen species when oysters were exposed to stressed conditions [77]. The adductor muscle can regulate the activities of enzymes involved in energy metabolism (especially anaerobic metabolism) and dynamics of free amino acids to achieve a metabolic homeostasis when oysters suffered from salinity stresses [78,79,80]. This functional specificity may underlie their distinctive transcriptional responses to hyper- and hypo-saline stresses. Considering the same genome shared among different tissues, evolutionary remodeling and regulation of gene expression is potentially the most crucial determinant for the generation of divergent responses across these tissue types [35,37,81].

LncRNAs are key regulatory elements that are linked to their specific subcellular localization, and expression, which function in *cis*- and *trans*-models at the transcriptional and post-transcriptional levels [41,82]. Interestingly, we found that the expression level of lncRNAs in oysters were substantially higher than those of protein-coding genes across all eight tissues and three salinity conditions. However, the traditional views revealed that lncRNAs are typically expressed at a lower abundance than mRNAs [82,83,84]. This inconsistency stressed that more lncRNAs are needed in oysters to regulate gene expression. In addition, most of environmentally responsive lncRNAs (DELRs) showed high tissue specificity that only 27.3–28.6% of DELRs were shared between striated muscle and hepatopancreas, while 35.6–44.9% of DEGs overlapped between these two tissues when exposed to salinity stresses. These finding demonstrated that the expression of lncRNAs is more restricted to specific tissues and conditions [41,82,83], which may regulate downstream gene expression. Recent genomic data revealed that lncRNAs can directly or indirectly (e.g., interact with transcription factors [85]) control the expression of nearby and distant genes and their functions can be of a structural/regulatory nature [49,51,86]. In this study, the expression level of genes (especially for environmental responsive genes: DEGs) associated with DELRs was significantly higher than that of genes unrelated to DELRs in both striated muscle and hepatopancreas in response to salinity stresses. Our results revealed that environmentally responsive lncRNAs facilitated the expression of associated genes to environmental challenges. Identical roles of some lncRNAs that positively enhance gene expression were also found in other species. These lncRNAs can be transcribed from enhancer or promoter regions when organisms were encountered with changes in temperature or other environmental factors [49,51,87]. We identified a pair of lncRNA-mRNA that exhibited positive regulatory relationship, where both the lncRNA (*MSTRG.46263.1*) and its associated salinity-responsive gene (*CREBL2*) were significantly increased under hyper-saline condition. The Hi-C loops and chromatin accessibility validated their direct interaction between *MSTRG.46263.1* and the upstream region of *CREBL2*, indicating *MSTRG.46263.1* may be an enhancer for *CREBL2*. Our findings provide genome-wide and individual evidence that lncRNAs control tissues-specific environmental responses through *cis*-acting regulation.

The Gene Ontology analysis of DEGs significantly associated with DELRs showed that there were tissue-specific divergent biological processes between striated muscle and hepatopancreas in response to hyper- and hypo-saline stresses. A common pathway (“response to stress”) related to environmental response/adaptation of marine species was enriched in both tissues (hepatopancreas under hyper-saline condition and striated muscle under hypo-saline condition), indicating they were exposed to severe environmental stresses [7,11,73,88]. This further supported our aforementioned arguments that the hepatopancreas was more sensitive to hyper-saline stress, while the striated muscle was more sensitive to hypo-saline stress. However, other classical environmentally responsive pathways [89,90,91,92,93] showed tissue-specific enrichment patterns in these two tissues. Genes responsible for programmed cell death were enriched in striated muscle, while genes responsible for cell–matrix adhesion and amino acid metabolism (glutamine and proline) were enriched in hepatopancreas [21]. This finding indicated that hepatopancreas still tried to recruit cellular resistant pathways under severe stress, while striated muscle suffered seriously cell death. Under temperately stress conditions (hepatopancreas under hypo-saline condition and striated muscle under hyper-saline condition), the hepatopancreas recruited osmoregulation of the pathway “water transport” to keep the physiological homeostasis via aquaporins (*AQPs*) [92,94]. The striated muscle activated the pathway of “mannan catabolic process”, which was identified as one of critical responses (controlling shell closure via muscle tension) to environmental changes in bivalve species [9]. Furthermore, we found that most genes in these pathways (*AQP8*, *STI1A*, *GADD45*, *HSP68*, *HSP70A*, *HSP70B*, *IAPs*, *STI1B*, *STI1C*, *PYCR1*, *GS*, *COQ7*, *β-mannanase*) exactly increased expression levels in response to salinity changes. These results demonstrated that different tissues exhibited divergent functional pathways to respond challenging conditions, even under the same stress. lncRNAs play an important role in mediating tissue-specific transcriptional responses to environmental changes. Further studies focusing on molecular mechanisms of tissue specificity for how lncRNAs regulate the certain pathways and the regulation of lncRNAs’ generation under changing environments are needed.

In conclusion, our results revealed that the Suminoe oyster has evolved tissue-specific genome-wide expression profiles in response to salinity stresses at both mRNA and lncRNA levels. This study provided new insights into lncRNA-mediated tissue-specific response capacity to environmental changes in marine invertebrates, and emphasized the significance of tissue-specific evaluations for the mechanisms and evolution of short-term contemporary responses and long-term evolutionary adaptation.

## 4. Materials and Methods

### 4.1. Oyster Samples

Wild oysters of *C. ariakensis*, ~60 mm in size and one year old [95], were collected from Taishan, China (21°95′ N, 122°83′ E). The oysters were transported to the mollusk laboratory at the Institute of Oceanology, Chinese Academy of Sciences in Qingdao. Oysters were cultured in 2 m × 2 m × 0.3 m aquaculture tanks containing aerated sand-filtered seawater maintained at 23.5–25.5 °C for one week. Commercial diatom powder was added as a food source, and the seawater was changed daily. Subsequently, oysters were randomly divided into three groups: one group was kept at their natural salinity conditions (20‰), one group was exposed to hypo-saline stress (0‰), and another group was exposed to hyper-saline stress (50‰) for 7 days. Freshwater from the tap was aerated for 24 h and used for stress experiments at a salinity of 0, as well as for preparing experimental water with a salinity of 20. Natural seawater combined with sea salt is used to prepare experimental water with a salinity of 50. The salinities were measured using a YSI 556 Multi-Probe System (YSI, Yellow Springs, OH, USA).

We sampled eight tissues (G, SmM, StM, MM, IM, heart, lip, and HE), using sterilized tweezers and scissors, of 3 oysters in each group on day 7. The samples were immediately frozen in liquid nitrogen and preserved in a −80 °C freezer for subsequent RNA extraction.

### 4.2. RNA-Sequencing

Total RNA was extracted from approximately 20 mg of the above 72 samples using TRIzol (Vazyme, Nanjing, China) reagent. The RNA concentration and quality were measured using a NanoDrop 2000 spectrophotometer (Thermo Fisher Scientific, Waltham, MA, USA) and 1.2% agarose gel electrophoresis. The VAHTS Universal V6 RNA-seq Library Prep Kit (Vazyme, Nanjing, China) was used to construct cDNA libraries, according to the manufacturer’s instructions. Then, the cDNA libraries were sequenced using BGISEQ-500 platform (BGI, Shenzhen, China), and 150 bp paired-end reads were generated.

Raw data were filtered to discarded low-quality reads, polyN-containing reads and adapter reads using fastp with default parameters [96], and obtained high-quality clean data. Then, the clean reads were mapped to the reference genomes of *C. ariakensis* [11] by HISAT2 [97]. Finally, the featureCounts (Version 2.0.6) [98] software was used to quantify gene expression levels, and transcripts per million (TPM) values for each gene were calculated using R4.3.1. A total of 500.19 GB of clean data was obtained and the average Q30 exceeded 95.0% (93.69–96.57%). The average mapping rate to the *C. ariakensis* reference genome was 64.04% (55.98–68.21%), covering 29,066 genes (electronic Appendix A).

### 4.3. Identification of lncRNAs

The transcript data of 72 samples were annotated and merged using StringTie (Version 2.2.1) [99] for subsequent identification of lncRNAs. Then, the GffCompare (Version 0.12.6) [100] program was applied to compare the merged transcript data with annotation data of *C. ariakensis*. The following steps were performed to screen unannotated transcripts that were defined as lncRNAs:

(1) Transcripts with class _ code of “i”, “u”, “x”, “j”, and “o” were selected;

(2) Transcripts with a length < 200 bp and an exon count <2 were removed;

(3) Transcripts with an FPKM (fragments per kilobase million) value > 0 were selected;

(4) The lncRNAs overlapping among the following four coding potential analyses were determined. We used CPC2 [101] (Coding Potential Calculator2), PLEK [102] (predictor of long non-coding RNAs and messenger RNAs based on an improved k-mer scheme) and CPAT [103] (Coding Potential Assessment Tool score < 0) to calculate the coding potential and only non-coding transcripts were retained for further analysis. The protein families database (Pfam [104]) was used to search for protein domains in the Pfam HMM library to identify transcripts with unknown protein domains. A total of 837 highly reliable lncRNAs with a FPKM > 0 were identified.

### 4.4. Differential Expression Analyses and Cluster Analysis

Differential expression analysis of lncRNAs and mRNAs of eight tissues was performed, respectively, using the DESeq2 package in R4.3.1. Genes or lncRNAs with |log2 (fold change)| ≥ 2 and *p*-value < 0.05 were identified as differentially expressed genes (DEGs) or lncRNAs (DELRs). Principal coordinates analysis (PCoA) was performed based on TPM for all the expressed genes and lncRNAs to show the differences among tissues under the same salinity conditions and differences within the same tissues under different salinity conditions.

### 4.5. Transcriptional Changes in Genes upon Salinity Stresses

For each tissue, the average TPM of each gene or lncRNA was calculated based on three replicates at each salinity condition. The plasticity was defined as absolute lg (fold change) of expression levels between salinity stress and native condition. Differences in plasticity were tested using the two-sample Wilcoxon rank-sum test using R4.3.1.

### 4.6. Construction of lncRNA-mRNA Co-Expression Network and Functional Enrichment Analyses

Pearson correlation coefficient tests were conducted to analyze the correlation among DELRs, DEGs, and non-DEGs (not differentially expressed genes) using R4.3.1. The result with |r| > 0.9 and *p*-value < 0.05 (to remove of false positives) were defined as significant correlation between the lncRNAs and mRNAs. Then, mRNAs were divided into four categories: DEGs significantly correlated with DELRs (scL-DEGs), DEGs not significantly correlated with DELRs (ncL-DEGs), non-DEGs significantly correlated with DELRs (scL-nDEGs), and non-DEGs not significantly correlated with DELRs (ncL-nDEGs). The co-expression network of lncRNA-mRNA was visualized using Cytoscape (Version 3.10.2) [105].

TopGO (Version 3.18) [106] software was used to perform enrichment analysis of the whole DEGs in lip, HE, StM, and SmM, as well as DEGs significantly correlated with DELRs in HE and StM with annotations from the Gene Ontology (GO) database.

### 4.7. High-Throughput Chromosome Conformation Capture (Hi-C) and Assay for Transposase-Accessible Chromatin Sequencing (ATAC-Seq)

The Hi-C and ATAC data of *C. ariakensis* were obtained from our previously published data [9]. Wild *C. ariakensis* oysters were collected from an estuary in the Bohai Sea near Binzhou [BZ] in northern China (38.18° N). Half of the oysters were kept in their native environment, and the other half were translocated to an estuary in the South China Sea near Qinzhou [QZ] (non-native southern environments) in southern China (21.74° N). After 2 months of out-planting, the adductor muscle was sampled from two oysters in each environment with significant salinity difference. Hi-C and ATAC-seq were conducted on the HiSeq X Ten PE150 platform (Illumina, San Diego, CA, USA). Details of sequencing and data processing were described in our previous study [9]. For two samples of oysters from the same location, two calling peaks were combined into one and visualized using Intergrated Genome Viewer (Version 2.19.1) [107].

## Figures and Tables

**Figure 1 ijms-26-04523-f001:**
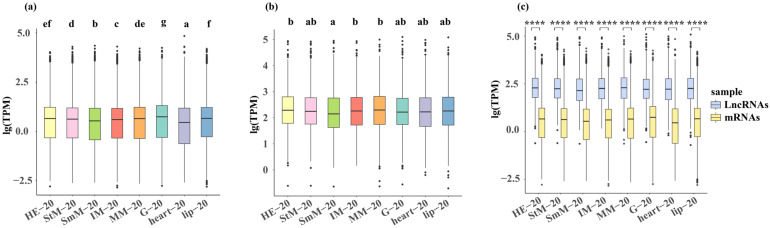
Genome-wide expression levels of mRNAs (**a**) and lncRNAs (**b**), along with their comparison (**c**) under the native salinity condition (20‰). Tissues are color-coded as: HE (light yellow), StM (pink), SmM (light green), IM (red), MM (orange), G (light blue), heart (purple), lip (blue). Letters and asterisks denote significant differences (*p* < 0.05).

**Figure 2 ijms-26-04523-f002:**
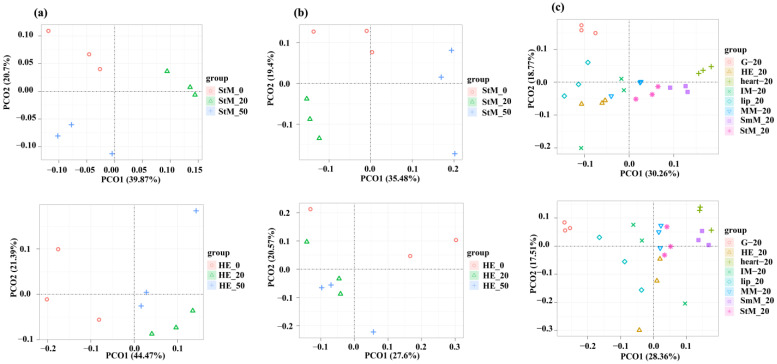
PCoA clustering of genome-wide expression profiles for mRNAs and lncRNAs. (**a**) Clustering of mRNAs in StM (**top**) and HE (**bottom**) under three salinity conditions (0‰, 20‰, 50‰). (**b**) Clustering of lncRNAs in StM (**top**) and HE (**bottom**) under three salinity conditions (0‰, 20‰, 50‰). (**c**) Clustering of mRNAs and lncRNAs across eight tissues under native salinity condition (20‰).

**Figure 3 ijms-26-04523-f003:**
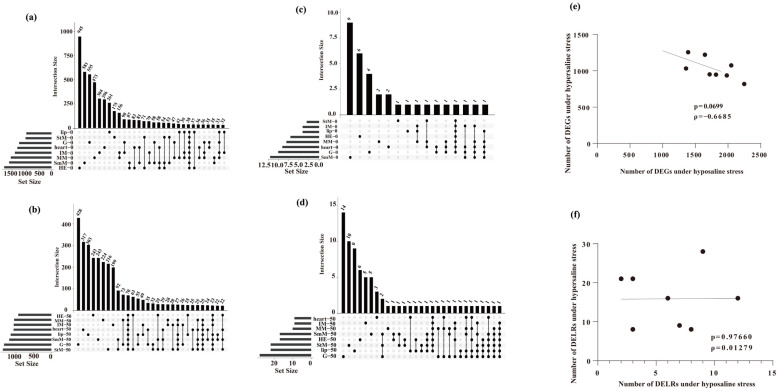
Number of DEGs (**a**,**b**) and DELRs (**c**,**d**) under hypo- (**up**) and hyper-saline (**bottom**) stresses. (**e**,**f**) Linear relationships of the number of DEGs (**e**) and DELRs (**f**) under hypo- (0‰) and hyper-saline (50‰) stresses across eight tissues. Each dot represents the DEG count of each sample under both conditions. The trend line indicates the correlation between these two variables.

**Figure 4 ijms-26-04523-f004:**
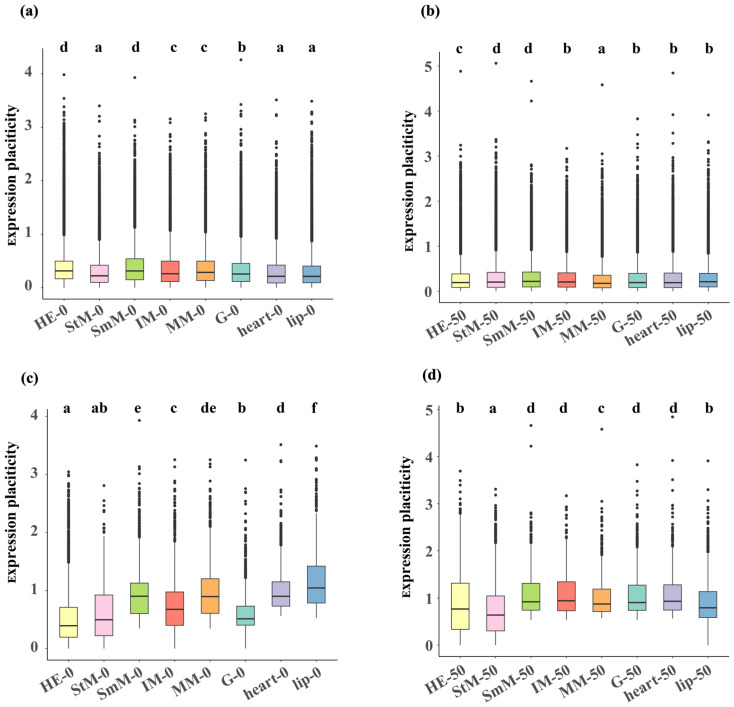
Expression plasticity of genome-wide mRNAs (**a**,**b**) and DEGs (**c**,**d**) under hypo- (**a**,**c**) and hyper-saline (**b**,**d**) stresses across eight tissues. Tissues are color-coded as: HE (light yellow), StM (pink), SmM (light green), IM (red), MM (orange), G (light blue), heart (purple), lip (blue). Letters indicate significant differences (*p* < 0.05).

**Figure 5 ijms-26-04523-f005:**
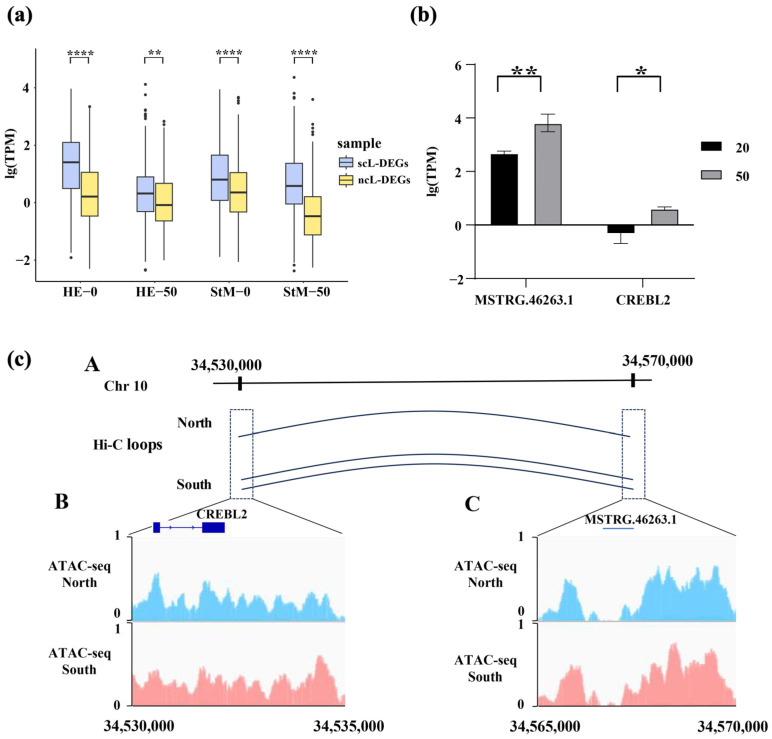
Expression levels of DEGs significantly correlated with DELRs (scL-DEGs) and those not correlated (ncL-DEGs) in HE and StM under salinity stresses (**a**), alongside a regulatory example involving mRNA (CREBL2) and lncRNA (MSTRG.46263.1) (**b**,**c**). (**b**) Expression levels of MSTRG.46263.1 and CREBL2 under native and hyper-salinity conditions. (**c**) Hi-C contact map between MSTRG.46263.1 and the 7 kb upstream region of CREBL2 (**A**). ATAC-seq tracks for CREBL2 and its 7 kb upstream region (**B**), as well as MSTRG.46263.1 (**C**), in oysters from native northern and non-native southern environments. Asterisks indicate significant differences (* *p* < 0.05, ** *p* < 0.01, **** *p* < 0.0001).

**Figure 6 ijms-26-04523-f006:**
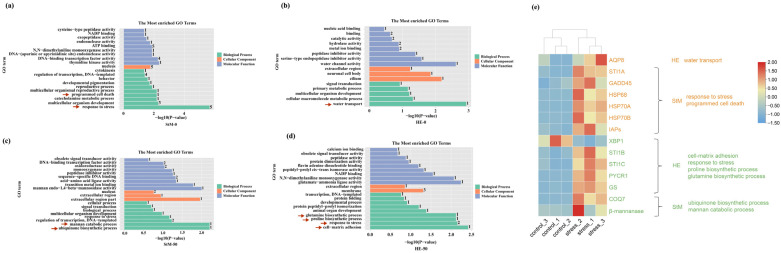
GO enrichment analysis for DEGs significantly correlated with DELRs in StM (**a**,**c**) and HE (**b**,**d**) under hypo- (**a**,**b**) and hyper- saline (**c**,**d**) stresses. Arrows represent the pathways relate to stress response. (**e**) Heat maps showing the expression levels of DEGs in significantly enriched GO terms, with orange indicating genes enriched under hypo-saline stress and green indicating genes enriched under hyper-saline stress.

**Table 1 ijms-26-04523-t001:** The number of DEGs significantly correlated with DELRs under hypo- and hyper-saline conditions.

	DELRs	DEGs
HE-0	6	767
HE-50	16	628
StM-0	2	336
StM-50	21	987

## Data Availability

The raw sequence reads have been deposited in the BioProject database (http://www.ncbi.nlm.nih.gov/bioproject) under accession no. PRJNA1194742 (accessed on 6 December 2024). Other figures and tables in this study have been uploaded as electronic Appendix A.

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
