# Peer review of "LncRNA-Mediated Tissue-Specific Plastic Responses to Salinity Changes in Oysters"

_ijms, 2025, doi:10.3390/ijms26104523_

Round 1

Reviewer 1 Report

Comments and Suggestions for Authors

This paper reported their work on tissue-specific responses to salinity changes by analysis of long non-coding RNA (lncRNA) and mRNA. The subject matter is of interest to the readers of the journal. The abstract is well written, but less supported by the results presented. The major concern is the lack of major results that they presented in which they failed to present specific data to support the title, nor the abstract. Specific comments are the following:

  1. The results section is empty that should be supported by data. Right now, 3.1 is a summary of the RNA-Seq data sets, which is really part of the Materials and methods. Although it is Ok to place it in the Results, but it is highly general.
  2. 2 described the principal component analysis, again, it is the method focused presentation, rather than what was really found.
  3. 3 is a core section of the results, i.e., DEGs and differentially expressed lncRNAs, but again, it is just lots of numbers without details. What kind of genes are induced or suppressed in the DEGs or DELR, and if the DELRs related to the DEGs, so what?

These questions are not answered.

  1. Throughout the Results section, it is apparent that they just presented the “framework”, but no detailed, making one wondering where did the details, or substance go?

In summary, the work is interesting, and the results, based on what they told the readers (without much data support) are significant, but the results are not presented. They are suggested to present the results as to what genes are induced, suppressed, how that is relevant to the salinity biology, what lncRNAs are induced or suppressed, what genes do they regulate, how those corresponding RNAs go up or go down, and such things are relevant to salinity responses and stress biology. These are significant, but they should have everything that is required to answer such questions.

Reviewer 2 Report

Comments and Suggestions for Authors

This is an interesting study which examine tissue-specific mRNA and lncRNA expression in response to hyper and hypo salinity stress. They found differential expression of mRNAs and lncRNAs across tissue types. They further focused on hepatopancreas and striated muscle tissue, examining the functions associated with differentially expressed genes and lncRNAs. This manuscript provides important insights into tissue-specific mRNA and lncRNA expression, which could help determine regulatory mechanisms of stress responses in oysters. The manuscript is clearly well written and objectives are very clear too. There was no major flaw in experimental design and results were interpreted too. If authors can address some of the points indicted below, this manuscript is strongly recommended for publication after relatively minor revision.

Introduction:

Line 51-52: The concept of phenotypic plasticity is not clearly defined, and it may confuse readers.

Line 112-114: The special and interesting traits of shellfish are their strong environmental adaptation and biomineralization abilities. Why only focus on the function of lncRNAs in environmental adaptation? How about their role in biomineralization?

Materials and methods:

Line 131: 600mm? Check the size

Line 134-137: Why choose 0‰ and 50‰ as the saline stress? Provide the reference about it.

Results: Relatively well written with high degree of clarity.

Discussion:

Line 361-362: If the statistical relationship is non-significant, how does it support the conclusion of an evolved trade-off strategy? More evidence or more discussion should provide for this support this point.

Line 384-385: Since hepatopancreas are related antioxidative capacity, is there any DGEs related to antioxidative capacity?

Line 448-449: What are the gens in these pathways, clarify it.

Reviewer 3 Report

Comments and Suggestions for Authors

Letter to Authors
ijms-3568521-v1
LncRNA-mediated tissue-specific plastic responses to salinity changes in the oyster
Mengshi Zhang, Jinlong Zhao, Ao Li, Mingjie Zhao, Meitong Huo, Jinhe Deng, Luping Wang, Wei Wang, Guofan Zhang, Li Li

250411

Dear authors,
This seems well-written MS (except for figures) dealing with transcriptional regulation/response to salinity changes in Suminoe oyster. Particular adaptation to salinity changes in this sessile estuarine animal is highly expected, and hence elucidation of the system/mechanism of the oyster from functional genomics viewpoint will make progress of understanding adaptation of animals in changing environments. It is thus worth publishing this MS in an international journal. Before publication it needs at least a round of major revision. Because the figure pictures are quite difficult to see, readers will hardly evaluate this MS. Result of north/south transfer is given without telling how. This is equivalent to "revision of experimental methods needed" that is a criterion of "major" in the journal. Several miscellaneous points also await revision. See below for detail. 

L18,49,128
contemporary ?
See L128. 

L24,99
lncRNA -> long non-coding RNA (lncRNA)
Spell-out at the first appearance mutually independent in the abstract and main text. 

L34 keywords
salinity response; oysters -> replace
Avoid listing words which appear also in the title. "specificity" and "plasticity" are in addition not very good. Duplicate hits upon computer search do not make sense. Give words that do not appear in the title to draw attention from wider readership. Posting words that neither appear in the abstract is better, because even in full-text search/indexing robots may not weigh much on words deeper (posterior) in the text. Hint: estuary, coastal ecosystem, tidal marsh, bivalve, sessile animal, shellfish, stress response, RNAi, transcriptome, RNA-seq, etc. 

L51
use avoidance (does not make sense) -> leave away

77
SNPs -> single nucleotide polymorphisms (SNPs)
3D -> three dimensional (3D)

L84
Tissue specificity of what?
Too much wide and general statement does not make sense. 

L99
H3K4me1/3 -> histone modification
3D

L101
long non-coding RNAs (lncRNAs) -> lncRNAs

L131
one year old ?
How did you identify their age?

L133
acclimated ?
How? What container? Running water? Salinity? Temperature?

L137-139
Oysters were cultured .. changed daily. ??
Inconsistent with the salinity manipulation. Move to L133?

L142
50.The -> 50. The
Check thoroughly for other typos. 

L144
(gill [G], smooth muscle [SmM], striated muscle [StM], marginal mantle [MM], inside mantle [IM], heart, lip, and hepatopancreas [HE]) -> (G, SmM, StM, MM, IM, heart, lip, and HE)
See L118. 

L151,153,155
Supplier information is needed. 

L176-178
The intersections .. as the putative lncRNAs. (redundant) -> delete
You may cite references omitted herein in L171,172,175. 

L201
Reference is needed for TopGO. 

L212
Reference is needed for IGV. 

L227,238,252,279,296,319 figure pictures
Fonts in the pictures are too small to see. They should at least as large as those in the main text. Do not make stupid proportional enlarging. Enlarge fonts only. 

L227
Order of tissues presented in a-c should be consistent. 

L234
figure 2a, b and figure 2a, b -> figure 2a, b and figure S2a, b ?

L238
Light-colored dots on PCA plains of figure 2 are also too small to see. Light-colored matters should be larger than darker ones for better visibility. 

L253
The number (does not make sense) -> Number
DEGs and DELRs -> DEGs (a, b) and DELRs (c, d)
Put necessary information ahead. 

L254
(a, b) -> delete
(c, d) -> delete
Later information does not make sense. 

L279
Order of tissues presented in a-d should be consistent. You do not mention ascending/descending order of expression plasticity across tissues but characteristics of particular tissues, and hence the current order makes it difficult for readers to find the particular tissues in the candlestick charts. 

L267 sub-section 3.4
You have defined the plasticity as fold change values in sub-section 2.5. Then, you better present those values in parentheses (ranges or arithmetic mean for example) to help readers' understanding. 

L287
andfigure ?

L309-311
Moreover, .. to non-native south environment (figure 5c).
Experimental setting must be presented in the M&M section. 

L336
divergent transcriptional profiles -> mRNA profiles ?
Make it contrasting. 

L339
respectively (verbose) -> delete
Self-explanatory words are dispensable. 

L342
also revealed in model species
Reference(s) is needed. 

L357,etc
strait -> striated
Check thoroughly. Or otherwise, you may use their acronyms in this section. 

L374-377
whereas induced .. upon hypo-saline stress. (wordy) -> revise
Avoid repeated words. Make abstract into noun phrases. I expect you could make compact less than 20 words from the current 37 words. 

L400-402
This inconsistence .. play critical role in regulating gene expression. ?
Is this logically correct? You may be correct, but an alternative in which each lncRNA copy has a weaker role for expression control (more copies needed) could also be possible. 
inconsistence -> inconsistency 

L422
suggesting .. may be (double weak wording) -> suggesting .. is .OR. indicating .. may be

L431
statement -> argument(s?)

L448
important role -> an important role

L454-457
Strait muscle .. through cis-model. (redundant) -> delete
A conclusion is not a summary. 

L487 references
Check the reference list carefully again from the beginning. Reference lists are frequently hotbeds of errors. You might add, omit or swap citation in the main text on the way internal revision. Numbering of the references might then shift. If so, readers think you are making irrelevant citation. It is the authors' responsibility that all references are properly cited.

Check thoroughly to make sure:
if separators of authors are a colon (L545,etc many),
if surnames come first for all authors (L649,etc),
if all authors are listed except for >10 in "et al." (L488,etc all),
if separator of authors from the paper/book titles is a dot (L488,etc all),
if journal titles are abbreviated when possible (L579,etc),
if abbreviated journal title words accompany a dot (L502,etc),
etc.
See the citation guide at:
http://www.mdpi.com/authors/references/

The following items may be helpful for further discussion. 

Alexa A, Rahnenfuhrer J, Lengauer T. 2006. Improved scoring of functional groups from gene expression data by decorrelating GO graph structure. Bioinformatics.22:1600-1607. 

Robinson JT, Thorvaldsdottir H, Winckler W, Guttman M, Lander ES, Getz G, Mesirov JP. 2011. Integrative Genomics Viewer. Nat Biotechnol 29:24-26. 

Comments on the Quality of English Language

Some typos and grammatical errors should be corrected.

Round 2

Reviewer 1 Report

Comments and Suggestions for Authors

My concerns are largely addressed.

Author Response

We sincerely appreciate your insightful suggestions throughout the review process

Reviewer 3 Report

Comments and Suggestions for Authors

Letter to Authors
ijms-3568521-v2
LncRNA-Mediated Tissue-Specific Plastic Responses to Salinity Changes in the Oyster
Mengshi Zhang, Jinlong Zhao, Ao Li, Mingjie Zhao, Meitong Huo, Jinhe Deng, Luping Wang, Wei Wang, Guofan Zhang, Li Li

250424

Dear authors,
This v2 MS is acceptable after correction of CARELESS ERRORS. 
Please note that there are very few people reading through your MS before publication. Some errors might be overlooked. There will be, on the other hand, many people who read your paper after it is published. Errors, if any, whatever trivial they were, can easily be found by some of the readers. Publishing a paper is a milestone of young scientists' research careers especially among your author group. BE EXTREMELY CAREFUL to check your MS from the beginning before publication to find and fix any errors. 

L396,etc (many)
straited -> striated
"STRIATED muscle" is translation of "Heng2-Wen2-Ji1". See L25,etc.
Check thoroughly.
Inconsistent and partly wrong wording clearly indicates different co-authors wrote earlier and latter parts, respectively, but it is the top and/or corresponding authors' responsibility that all parts are properly and consistently written.

Comments on the Quality of English Language

Inconsistent and partly wrong wording clearly indicates different co-authors wrote earlier and latter parts, respectively, but it is the top and/or corresponding authors' responsibility that all parts are properly and consistently written.
